# Inhibition of Primary Photosynthesis in Desiccating Antarctic Lichens Differing in Their Photobionts, Thallus Morphology, and Spectral Properties

**DOI:** 10.3390/microorganisms9040818

**Published:** 2021-04-13

**Authors:** Miloš Barták, Josef Hájek, Alla Orekhova, Johana Villagra, Catalina Marín, Götz Palfner, Angélica Casanova-Katny

**Affiliations:** 1Department of Experimental Biology, Faculty of Science, Masaryk University, Kamenice 5, Building A13/119, 625 00 Brno, Czech Republic; mbartak@sci.muni.cz (M.B.); jhajek@sci.muni.cz (J.H.); a.orechova@seznam.cz (A.O.); 2Laboratory of Plant Ecophysiology, Faculty of Natural Resources, Campus Luis Rivas del Canto, Catholic University of Temuco, Rudecindo Ortega #03694, 4780000 Temuco, Chile; jovyvillagra@gmail.com; 3Laboratory of Mycology and Mycorrhiza, Faculty of Natural Sciences and Oceanography, Campus Concepción, Concepción University, 4030000 Concepción, Chile; catmarin@udec.cl (C.M.); gpalfner@udec.cl (G.P.)

**Keywords:** maritime antarctica, King George Island, lichen dehydration, chlorophyll fluorescence, stress tolerance

## Abstract

Five macrolichens of different thallus morphology from Antarctica (King George Island) were used for this ecophysiological study. The effect of thallus desiccation on primary photosynthetic processes was examined. We investigated the lichens’ responses to the relative water content (RWC) in their thalli during the transition from a wet (RWC of 100%) to a dry state (RWC of 0%). The slow Kautsky kinetics of chlorophyll fluorescence (ChlF) that was recorded during controlled dehydration (RWC decreased from 100 to 0%) and supplemented with a quenching analysis revealed a polyphasic species-specific response of variable fluorescence. The changes in ChlF at a steady state (Fs), potential and effective quantum yields of photosystem II (F_V_/F_M_, Φ_PSII_), and nonphotochemical quenching (NPQ) reflected a desiccation-induced inhibition of the photosynthetic processes. The dehydration-dependent fall in F_V_/F_M_ and Φ_PSII_ was species-specific, starting at an RWC range of 22–32%. The critical RWC for Φ_PSII_ was below 5%. The changes indicated the involvement of protective mechanisms in the chloroplastic apparatus of lichen photobionts at RWCs of below 20%. In both the wet and dry states, the spectral reflectance curves (SRC) (wavelength 400–800 nm) and indices (NDVI, PRI) of the studied lichen species were measured. Black *Himantormia lugubris* showed no difference in the SRCs between wet and dry state. Other lichens showed a higher reflectance in the dry state compared to the wet state. The lichen morphology and anatomy data, together with the ChlF and spectral reflectance data, are discussed in relation to its potential for ecophysiological studies in Antarctic lichens.

## 1. Introduction

The lichen biota of the Antarctic continent dominates the polar tundra with more than 350 species distributed throughout the few small ice-free areas [1,2]. Temperature and water are the two principal factors limiting the distribution of living organisms in Antarctica [3]. Both lichens and mosses penetrate further into the discontinuous small ice-free microsites that offer colonizable substrates, reaching almost 86 °S in the case of lichens [4]. Although almost the entire distribution is modeled based on the temperatures that make life feasible, little is known about the role of water in survival on the frozen continent. However, for microenvironments, it has been reported that beyond 72 °S, the most determining factors of diversity are the length of the active period and the available water, which is closely linked to temperature [5]. Various studies have shown that lichens are resistant to low temperatures and present positive photosynthesis at temperatures close to 0 °C [6]. However, it has also been recorded that the optimum temperature for net photosynthesis for several lichens is between 5–15 °C [7]. Furthermore, it is expected that, as global warming progresses, these organisms will be affected by the increase in temperature [8], which in Antarctica is as high as approximately 3 °C, although this varies throughout the continent. On the other hand, the majority of Antarctic lichens are saxicolous because rocks and stones are the most available substrate on the Antarctic continent; therefore, the water content variations to which they are exposed are greater because the rocks do not retain water. In this context, although their response to temperatures is well known, less is known about the response of Antarctic lichens to a lack of water in the field. Lichens are poikilohydric organisms that obtain water from atmospheric humidity or from rain. In Antarctica, where it rarely rains, water is mostly available in liquid form from the water runoff from snowbanks and glaciers or from snowfall in summer. Therefore, during the study of the in situ response of the crustose lichen *Placopsis antartica*, it was found that the dominant factor that affected the electron transport rate (ETR) was the speed with which the thallus moistened in comparison to the controls [9]. Hence, from the ecological perspective, it is important to investigate whether Antarctic lichens present differences in their ability to respond to changes in humidity, as well as whether there are differences between species displaying different morphotypes [10,11]. During the Antarctic summer, lichens absorb water from rain, mist, and thawing snow [12]. However, in Continental Antarctica, lichens can also absorb water from snow sublimation [13].

Lichens are desiccation-tolerant organisms that undergo a large number of dehydration/rehydration cycles during their lifetime [5]. When lichens undergo desiccation from a wet to a dry state, they lose their photosynthetic activity and gradually become physiologically and photosynthetically inactive. For severely desiccated thalli, CO_2_ exchange is ceased in lichen photobionts until they become hydrated again [7]. Desiccation in lichens correlates with a decline in primary photosynthetic activity, a loss of variable chlorophyll fluorescence (ChlF), and a decrease in their overall fluorescence yield. These changes are accompanied by a decoupling of photosystems I and II in the algal/cyanobacterial photobiont during thallus desiccation. Lichens activate several protective mechanisms during desiccation, which decrease ChlF. The mechanisms quench ChlF and protect the chloroplastic apparatus from over-energization during desiccation. The changes enable photobiont cells and their chloroplasts to keep functioning, even under severe desiccation. The underlying mechanisms have been studied over the last few decades, and several quenching mechanisms have been identified in lichens [14]. Among relevant studies, energy dissipation from desiccating mosses and lichens was described by [15,16,17,18]. The mechanism comprises effective quenching centers that appear during desiccation and act as thermal dissipation units. 

This study focuses on the effects of lichen desiccation on the shape of slow Kautsky kinetics (KKs) of ChlF measured in dark-adapted samples exposed to continuous light. The KKs show a rise from the O (background ChlF) to the P peak, followed by a polyphasic SMT phase. The OPSMT transient reflects important points on KKs, where O stands for the origin, P for peak, S for a semi-steady-state ChlF level, M for local maximum, and T for a terminal steady-state ChlF level. The OPSMT shapes have been analyzed in detail by Riznichenko et al. (1996) [19], Govindjee (see [20,21]). The authors described the ChlF signal change within the PSMT part of the KKs and attributed it to the photochemical and nonphotochemical processes that take place in a photosynthetic apparatus. Regarding the photochemical processes, the reoxidation of reduced Q_A_ due to the photosynthetic electron transport chain causes the decrease in the PSMT phase. However, other processes, such as those involved in nonphotochemical quenching, could also be involved (see below). Generally, the SMT phase of the ChlF signal forming the induction curve and the appearance of additional maxima are caused by stimulating the dark reactions of the Calvin-Benson cycle of the CO_2_ fixation (see [22]). The SMT part is polyphasic due to several co-acting processes. Nonphotochemical fluorescence quenching (NPQ) results from these processes as the formation of the transthylakoidal proton gradient [23], phosphorylation of the light-harvesting complex (see [24]), oxidation of the plastoquinone pool, and photoinhibition [25]. The P-S-M transition is denoted to the components of nonphotochemical quenching (qN), i.e., energy-dependent quenching (qE), state transition quenching (qT), and photoinhibitory treatment (qI) [26]. For qE, several underlying mechanisms have been proposed, such as quenching in light-harvesting complexes [27]. For qT, the “spillover” of excitation energy from PSII to PSI has been considered; however, the involvement of excitation spillover during the P-S phase is improbable [28]. For the S-M phase, the involvement of the state transition was reported previously [29]. Although qI’s involvement in the SMT phase is minor, photoinhibition-induced changes and the activation of photoprotective mechanisms must be taken into consideration. Because a proportion of the qN components are species- and treatment-specific, it may be suggested that the entire PSMT chlorophyll fluorescence transient reflects a superimposition of several processes.

Compared to the numerous studies that exploit KKs to analyze stress on PS II functioning, KKs has only been used sporadically in lichens. It was used to characterize their sensitivity to photoinhibition [30] and freezing stress [31]. Nabe et al. (2007) [32] studied the sorbitol effect on the KKs shape in liverwort (*Marchantia polymorpha*) and moss (*Bryum argenteum*). In this study, we focused on the PSMT phase of the KKs shape. We hypothesized that the KKs would be species-specific and sensitive to the desiccation of selected Antarctic lichens. Therefore, we measured the KKs of selected lichen species and analyzed the parameters derived from the O, P, S, M, and T ChlF signals (e.g., the ratios related to peak ChlF [P]), such as P/M, P/T, and others. We hypothesized that they would be related to the relevant water content (RWC) in thalli.

A dehydration-induced decrease in variable chlorophyll fluorescence that results in a decrease in slow ChlF transient has been shown several times for lichens. The decrease in O, P, S, and T ChlF signals is accompanied by the inhibition of primary photosynthetic processes, which is demonstrated by a decrease in potential yield of photochemical processes in photosystem II (F_V_/F_M_), and effective quantum yield of photosystem II (Φ_PSII_). Such response has been found in cyano- and chlorolichens (see [33]). However, the decline is species-specific and bi- or triphasic if the supersaturation effect takes place. The latter is true for some cyanolichens and *Nostoc commune* colonies [34]. Therefore, in our study, we expected species-specific responses to the ChlF parameters when evaluating the photosystem II activity of lichen photobiont, such as F_V_/F_M_, Φ_PSII_, and steady-state chlorophyll fluorescence (F_s_) in gradually desiccating lichen samples. Additionally, we were interested in the RWC at which the first signs of inhibition of the photosynthetic processes in PSII appear and the RWC at which half of the maximum ChlF parameters are found. Emphasis was also given to the critical RWC, i.e., the RWC at which the individual species show a full limitation of the primary photochemical processes of photosynthesis. Furthermore, we expected species-specific involvement of NPQ, which is considered a protective mechanism that is activated during desiccation in lichens [35]. We evaluated species-specific sensitivity of primary photosynthetic processes in several chlorolichens at low desiccation. This study is a follow-up study of a previous one [33], which focused on desiccation-induced limitation of primary photosynthetic processes in lichens monitored by ChlF parameters (Φ_PSII_ and nonphotochemical quenching of absorbed light energy—qN) and changes in spectral reflectance indices during thallus desiccation. In this study, we supplemented the approach by using a more detailed analysis of the shape of slow Kautsky kinetics during desiccation. We hypothesized that Antarctic lichens from King George Island, specifically their primary processes of photosynthesis, will be highly resistant to desiccation.

## 2. Material and Methods

### 2.1. Site Description, Lichen Species Collection, and Handling

The lichen material was collected during the Chilean Antarctic Expedition (ECA56) on the Fildes Peninsula (62°12′25″ S, 58°58′26″ O), King George Island (KGI), South Shetland Island Archipelago (Figure 1), which is located close to the northern Antarctic Peninsula. For terrestrial vegetation, the greatest diversity, cover and growth rates are contributed by cryptogam species in the northern part of the peninsula up to 72° S, where the factor that best correlates with the exuberance of the polar tundra would be the annual temperature [4]. Therefore, the South Shetland Archipelago in the maritime Antarctic is considered a diversity hotspot. Fildes Peninsula is the second largest ice-free area within the Archipelago, with a mean annual air temperature at sea level of −2.3 °C at Bellinghausen Station, and the annual decadal air temperature trend between 1969–2010 being about 0.259 K/decade [36]. The tundra vegetation consists of expanding, well-developed lichen and moss communities. A total of 61 moss species have been documented on King George Island, of which 40 are present on Fildes Peninsula [37], with also about 109 lichen species being distributed along the ice-free areas [38]. The lichen samples were transferred to Professor Julio at Escudero Station from Instituto Antártico Chileno (INACH) in January 2019, and the voucher specimens were stored at the Fungarium of the Universidad de Concepción, CONC-F.

### 2.2. Species Characteristics

For the study, we selected the following five Antarctic lichens (Figure 2 and Figure 3) that differ in their thallus morphology, photobiont, and morphotype: foliose (*Parmelia saxatilis*), crustose (*Placopsis antarctica*), and fruticose species (*Himantormia lugubris*, *Ramalina terebrata*, and *Lecania brialmontii*)*,* as shown in Table 1.

To study the anatomy of the lichen thalli, freehand cross-sections were made using a razor blade; thickness of the upper and lower cortex, medulla, and algae layer were measured in only two species, *Himantormia lugubris* and *Parmelia saxatilis*, due to the restrictions imposed by the quarantine. Lichen samples were put on glass slides before being observed under a Leitz Dialux microscope (Leitz, Wetzlar, Germany) at 100× magnification and documented using a Nikon Coolpix 950 digital camera (Nikon, Tokyo, Japan) attached to the microscope.

### 2.3. RWC during Dehydration

After collection in the field, the thalli of the lichen species chosen for the experiments were fully hydrated at 15 °C for 24 h in closed Petri dishes until the maximum weight was reached (this was tested by weighing the lichen on a laboratory analytical scale [Brand, Adam Equipment, Oxford, MS, USA]). The fully hydrated thalli were then dried at room temperature (18 °C, 40% RH) in the laboratory at the Escudero Station (Fildes Peninsula). During desiccation, the thalli were regularly (typically in around ten-minute intervals) weighed to evaluate the RWC which was calculated using the following equation: RWC (%) = [(Fw − Dw)/(Ww − Dw)] × 100, where Fw is the actual fresh weight of a sample, Dw is the weight of the fully dry sample (oven-dried sample at 35 °C for 24 h), and Ww is the weight of the fully hydrated sample. The weighing of the thalli and RWC evaluation lasted until a constant weight of dry thalli was reached.

### 2.4. Chlorophyll Fluorescence Measurements

The samples were collected and immediately remoistened for 24 h under the natural outside temperature. Before the dehydration measurements, the samples were tested for F_V_/F_M_ after 24 h, 25 h, and 26 h of rehydration. When F_V_/F_M_ reached a maximum and constant value, the sample was considered vigorous and the primary photosynthetic processes fully activated. From the fully wet (RWC = 100%) to dry (RWC = 0–10%) states of the studied species, the ChlF parameters were measured repeatedly, typically in 40-min intervals.

Laboratory measurements using the slow Kautsky kinetics (KKs) method supplemented with saturation pulses in dark- and light-adapted states were used. The chlorophyll fluorescence parameters were recorded using a FluorCam HFC 1000-H (Photon Systems Instruments, Drásov, Czech Republic) and the FluorCam v. 7.0 software. Additionally, Kautsky kinetics supplemented with a quenching analysis was used. The method starts with a saturation pulse applied in a dark-adapted state (ten minutes) to induce the maximum ChlF (F_M_) followed by ten seconds of dark. Then, the samples were exposed to actinic light (100 μmol (photons) m^−2^ s^−1^) for 300 s, and a polyphasic time course of the ChlF emission was recorded. When a steady-state ChlF was reached (after 300 s), another saturation pulse was applied to induce F_M_’ levels of ChlF, i.e., the maximum ChlF value in light-adapted material. After switching off the actinic light, background ChlF (F_0′_) was recorded for 20 s. Standard ChlF parameters (F_V_/F_M_, Φ_PSII_, NPQ, F_s_) were calculated using FluorCam software. Their dependence on RWC is presented in this study. For a more detailed analysis, the dehydration response curves of the effective quantum yield of the photosynthetic processes in PSII (Φ_PSII_) and the steady-state chlorophyll fluorescence (Fs) were selected. The critical points were distinguished for the Φ_PSII_ and Fs dehydration response curves, which denoted the RWC at which the ChlF parameter was limited to 0.

The dehydration response curves of Φ_PSII_, NPQ, and F_S_ were plotted and analyzed. Species-specific responses in the dehydration-induced decline in photosynthetic parameters were noted, and the RWC at which the functional changes occurred were evaluated.

The records of ChlF transients for particular lichen species and RWC were analyzed. On the slow Kautsky kinetics of ChlF, the levels O, P, S, M, and T were identified, along with the times at which they were reached. This was done using FluorCam software after 100× magnification of the curve (y axis: ChlF as dependent variable), which distinguished the particular ChlF levels, even on seemingly flat curves. Particular species-specific levels (P, S, M, T) were found by the software, as the ChlF signals reached at the times of 2.0 s (for P) 4.8–5.2 s (for S), 16.0–28.6 s (for M), and 300 s (for T) after the continuous light inducing KKs was switched on. The exception was *P. antarctica* (algal part) where S and M ChlF levels were found in 16.2 and 50.0 s. The effects of RWC on the above-specified ChlF levels and ratio parameters (P/S, P/M, S/M, M/T) were then evaluated according to [40].

### 2.5. Spectral Properties in the Wet and Dry States of Thallus

Reflectance spectra within the range of 380–800 nm were measured using non-imaging spectro-reflectometers, PolyPen RP 400 (UV-VIS, Photon Systems Instruments, Brno, Czech Republic). The measurements were conducted in the wet and dry states of the thallus to evaluate the hydration-dependent changes in spectral reflectance curves and derived from spectral indices (see Table 2). Lichen thalli were placed into a clip in the PolyPen’s measuring head, which allowed a constant distance between the detector and the lichen. In the clip, a short darkening period (approx. one minute) was allowed before single measurements of the spectral reflectance were taken. After downloading the files from the spectro-reflectometer, the mean ’wet’ and ’dry’ spectrums were calculated. The means of particular spectral reflectance indices were evaluated, and the change between the dry and wet states was discussed.

### 2.6. Statistical Analysis

Unless stated otherwise, the statistical analysis was done by an ANOVA test (RWC effect on ChlF parameters) and Student *t*-test (spectral reflectance curves), with statistically significant differences of *p* ≤ 0.05.

## 3. Results

### 3.1. Anatomy of the Species

A microscopy study revealed species-specific differences in the heteromerous thallus morphology and anatomy of the five different Antarctic lichens. Qualitative differences were found in the four thalline layers, upper cortex, green-algae layer (Trebouxioid), medulla, and lower cortex. In *P. saxatilis*, the algal layer is very thin and continuous, and the thallus thickness ranges between 152–159 μm (Figure 2a,b). For the fruticolose *Himantormia lugubris*, a very thin algal layer (compared to the thallus thickness) is located beneath the cortex (Figure 2c,d). However, the algal cells do not form an evenly thick layer but small clusters with some substantial spaces (up to 15 μm) between them (not shown). The upper cortex averages 15 μm, and thallus thickness fluctuates between 168–372.0 μm. *Lecania brialmontii* is a microfruticolose lichen that forms a pulvinate cushion. The thallus shows terete ramification below the upper cortex with a continuous algal layer, which can form between 25–30% of the thallus (Figure 2e,f). *Placopsis antarctica* has two types of photobionts—the green algal layer below the upper cortex (Figure 3a,b) and the cephalodia in the center of the thallus with cyanobacteria (Figure 3a,c). For *R. terebrata*, the old thallus parts exhibit an algal layer of varying thickness that is located beneath the cortex (Figure 3d,e). Young tips show an irregular distribution of the algal photobiont within a thallus cross-section. The algae form a series of cell clusters with a patchy distribution.

### 3.2. Dehydration Response Curves of the Potential and Effective Quantum Yield of PSII

Most species showed no limitations of F_V_/F_M_ (Figure 4) and Φ_PSII_ (Figure 5) in the thalli desiccating from a wet state (RWC: 100%) to a semi-dry state (RWC: approx. 35%). The cephalodium of *P. antarctica* and *L. brialmontii* had a different dehydration response curve with a slight but constant increase in F_V_/F_M_ and Φ_PSII_ when the RWC decreased from 100% to 20% (for F_V_/F_M_) and 100% to 30% (for Φ_PSII_), see the inset in Figure 4 and Figure 5. Then, with further desiccation below the RWC of 30%, a significant decline in the F_V_/F_M_ and Φ_PSII_ was found with further thallus desiccation at RWCs below 25%. Species-specific dehydration-dependent fall in F_V_/F_M_ and Φ_PSII_ started at an RWC range of 22–32%. With further desiccation, species-specific differences were more distinguishable. This was particularly true for the RWC_1/2_ values, in which F_V_/F_M_ and Φ_PSII_ reached their half maximum values. For F_V_/F_M_, RWC_1/2_ was found below 15% and declined in the following order: *Ramalina terebrata* (11.80%), *Himantormia lugubris* (9.72%), *Lecania brialmontii* (9.70%), *Parmelia saxatilis* (8.90%), algal part of *Placopsis antarctica* (8.62%), and cephalodium of *P. antarctica* (2.01%). For Φ_PSII_, although the species order of RWC_1/2_ differed, it generally had higher RWC_1/2_ values than F_V_/F_M_, as follows: algal part of *P. antarctica* (19.4%), *P. saxatilis* (16.7%), *R. terebrata* and cephalodium of *P. antarctica* (11.80%), *L. brialmontii* (11.4%), and *H. lugubris* (10.2%). Generally, low values were found for the critical RWC (RWC_crit_), at which F_V_/F_M_ and Φ_PSII_ reached zero. In all species, the RWC_crit_ was below 5%. For F_V_/F_M_, the values reached 0.85 % (*P. saxatilis*), 0.25 % (*H. lugubris*), 1.20% (*L. brialmontii*), 0.95% (*P. antarctica*—algal part), 0.20% (*P. antarctica*—cephalodium), 1.05 % (*R. terebrata*). For Φ_PSII_, the RWC_crit_ values reached 8.2% (*P. saxatilis*), 3.3% (*H. lugubris*), 4.4% (*L. brialmontii*), 5.0% (*P. antarctica*—algal part), 4.3% (*P. antarctica*—cephalodium), and 2.9% (*R. terebrata*). For the dehydration response curve of *P. antarctica*, a significantly higher Φ_PSII_ was found for the algal part of the thallus than for the cephalodia with the RWC declining from 100 to 15 %. In final phase of desiccation (RWC below 10%), however, Φ_PSII_ values are almost identical in algal and cyanobacterial parts of the thallus, contrasting with F_V_/F_M_, where higher values are found for cephalodia than for the algal part. With the exception of cephalodium of *P. antarctica* (F_V_/F_M_) and *H. lugubris* (Φ_PSII_,), the values of F_V_/F_M_ and Φ_PSII_ were highly related to steady state ChlF (Fs), see Table 3.

Steady-state chlorophyll fluorescence (Fs) declined in a polyphasic manner with ongoing thalli dehydration in all species (Figure 6). It showed a slight decline in the RWC range, as it decreased from 100% to 30%. Then, like F_V_/F_M_ and Φ_PSII_, Fs started to decline more significantly at a RWC of about 30%. Except for *H. lugubris* and cephalodium of *P. antarctica*, Φ_PSII_ correlated with Fs (R^2^ over 0.97), which indicates a high potential for the Fs signal to monitor vigor and photosynthetic activity in *P. saxatilis*, *L. brialmontii*, *R. terebrata*, and *P. antarctica* (algal part).

NPQ increased with desiccation, more apparently at the RWCs below 20%. The dehydration curves were of similar shape with the exception of *L. brialmontii* and *H. lugubris.* The two species showed smaller NPQ increase in the RWC declining from 20 to 0% RWC than the other ones.

### 3.3. Slow Kautsky Kinetics Recorded for the Lichens at Different RWCs during Desiccation from Fully a Wet to a Dry State

The chlorophyll fluorescence signal decreased with thallus desiccation in all species except for *H. lugubris*, in which the ChlF signal showed an increase in RWC, followed by a decrease from 100% to 60%. In *L.*
*brialmontii*, the KKs was almost identical in the RWC range during desiccation from 100% to 20%. However, in all species, a significant ChlF decrease in the KKs was apparent at RWCs below 20%. The desiccation-induced decrease was demonstrated by a “flattening” of the slow KKs and a decrease in the P, S, M, and T ChlF values (Figure 7). Similarly, a decrease was found for the ChlF values that were reached after the application of saturation pulses (F_M_ data not shown in Figure 7). These changes were attributed to the generally increased nonphotochemical quenching and the changes in the optical properties of lichen thalli during desiccation (for more details see Discussion). Additionally, some species-specific changes were observed in the KKs shape, specifically for the ChlF fluorescence signals reached at particular O, P, S, M, and T levels (Figure 7, see the upper left and lower left panels): (1) F_P_ < F_M_ with the S point becoming less distinguishable with pronounced desiccation (in this case, F_M_ does not denote to maximum ChlF reached after saturation pulse applied to a dark adapted sample but ChlF level reached at M in Figure 7), (2) time at which the M point was reached (e.g., 36 s in *R. terebrata*, 66 s in *P. saxatilis*). Therefore, desiccation induced some minor changes in the index parameters (P/S, P/M, S/M, M/T). These were small but apparent in all species except for *L. brialmontii* (see Table 4), which showed the KK’s “plateau type,” which is typical for most cyanobacteria [43]. The other species showed the typical OPSMT shape for chlorolichens.

### 3.4. Analysis of Reflectance Spectra in Dry and Wet States

The spectral reflectance curves, PRI, and NDVI showed species-specific sensitivity to dehydration (see Table 5, Figure 8). *H. lugubris* did not show any change between the spectrum recorded in dry and wet states, while the *Lecania brialmontii* spectra showed a lowering of reflectance from a dry to a wet state throughout the whole wavelength interval. The reflectance spectrum of dry *L. brialmontii* increased almost linearly between 300 nm to 680 nm. However, a local peak was apparent at 640 nm, followed by a red-edge increase starting at 680 nm. In both the wet and dry states, the red-edge increase was biphasic with a faster increase in the wavelength range of 680–720 nm, followed by a slower increase within the range of 720–780 nm.

NDVI and PRI differed for dry and fully hydrated thalli of the experimental lichen species. However, the response was species-specific. NDVI decreased in the hydrated state for *H. lugubris* and *P. saxatilis* but increased in *L. brialmontii*, *R. terebrata* and *P. antarctica*. However, in wet states, the values of PRI either increased (*H. lugubris*), decreased (*P. saxatilis*, *P. antarctica*), or showed no change (*L. brialmontii*, *R. terebrata*).

## 4. Discussion

### 4.1. Chlorophyll Fluorescence Parameters during Desiccation

The results show that Antarctic lichens have a high desiccation tolerance because all species showed a decline in F_V_/F_M_ and Φ_PSII_ at RWCs below 20% to 30%, as has been reported previously for *Cladonia borealis* [44]. Recent studies support the idea that liquid water availability is the main limiting factor for lichen photosynthesis (see [45]). For all experimental species, the values of F_V_/F_M_ and Φ_PSII_ were fairly constant in the thalli, desiccating from 100% to 30% RWC (Figure 4 and Figure 5). Except for *L. bialmontii* and *Nostoc*-containing cephalodium of *P. antarctica* (see the insets in Figure 4 and Figure 5), the F_V_/F_M_ and Φ_PSII_ values showed either no change or a decrease when the RWC declined from 100% to 30%. The slight increase of F_V_/F_M_ and Φ_PSII_ in *L. brialmontii* and cephalodium of *P. antarctica* that was found with desiccation from 100% to 30% of RWC can be attributed to the limited CO_2_ diffusion into the thallus in the wet state. This phenomenon is caused by exopolysaccharidic envelopes of cyanobacteria cells, which in the wet state represent a physical barrier for CO_2_ transfer. Therefore, photosynthetic processes are limited in fully hydrated thalli and increase with partial dehydration, as shown by *Nostoc commune* colonies [34].

Considering the microsite conditions at King George Island and thallus morphology, the crustose *Placopsis antarctica*, which grows on stones on the ground in a humid environment, passes faster desiccation during the summer season (shown by [9]) than *Lecania brialmontii*, which grows on coastal rocks with water runoff and high humidity. *L. brialmontii* is a microfruticulose lichen with a pulvinate cushion form where water can be retained for a long time. This has been found in similar morphological arrangements in Antarctic cushion mosses [46], where individual specimens lose water faster than the cushion form. A minor decrease in F_V_/F_M_ and Φ_PSII_ was found in *R. terebrata*, which grows on higher rocks that are exposed to the wind. On the other hand, *H. lugubris* is the species that showed the highest tolerance to drought, as indicated by higher values of Φ_PSII_ data points at RWCs below 10% than for the rest of species (Figure 5) and generally low NPQ values at the same RWC interval (Figure 6). The species grows on stones in drier areas to form extensive communities with other fruticose lichens [47]. The behavior of *H. lugubris* is interesting because the KKs analysis showed that the ChlF signal is most effective at a lower RWC (see below). The maximum value of Φ_PSII_ found for *H. lugubris* was comparable to the value reported by [47] for grey thallus branches.

A significant S-curve decrease of F_V_/F_M_ and Φ_PSII_ at RWCs of below 20–30% was found for the five experimental species (Figure 4 and Figure 5) and has been well documented for a variety of lichens [48,49]. Similarly, an earlier study [50] reported a rapid decline in F_V_/F_M_ in lichens when the RWC decreased below 20%. These changes indicate a severe limitation of PS II in lichen thalli in the final stage of desiccation. F_V_/F_M_ reached 10% of their maximum value at the RWC values below 10%. Such values are comparable to the data reported previously [51] for three lichen species from Norway. This suggests that lichens are capable of performing primary photochemical processes of photosynthesis at low thallus hydration.

During the desiccation of the lichen thallus, excess ROS formation occurs in PS II and other chloroplastic/cellular compartments of a photobiont. Desiccation-induced ROS formation in PS II results in high PS II pressure and, consequently, in photooxidative injury of pigment-protein complexes of PSII. These changes lead to a reduction of photosynthetic efficiency (F_V_/F_M_ and Φ_PSII_ decline, see Figure 4 and Figure 5). High levels of ROS are harmful not only to PS II but also to essential biomolecules, including nucleic acids, proteins, and lipids [52]. Moreover, the lack of water molecules in desiccating lichens combined with sunlight may overexcite the RCs of PS II, regardless of low lighting [53]. These changes are accompanied by an NPQ increase in RWCs below 20% (see Figure 6), which comprises several protective mechanisms that enhance desiccation tolerance. The reason behind the lower NPQ in *L. brialmontii* at the RWC below 20% than in other species is unknown. Such low activation of NPQ, however, suggests yet unidentified photosynthetic peculiarities of the *L. brialmontii* photobiont. NPQ has at least the following three components: (1) pH-dependent energy dissipation in the antenna system of PSII (qE); (2) a state transition between PSII and PSI (qT); and (3) a photo inhibitory quenching (qI) (see [54]). Our ChlF measurements could not determine which of the three components played a major role in the response of the experimental species to thallus dehydration. However, similarly to [44]—Cho et al. (2020), we suggest that qE could contribute to a rapid reduction in PS II excitation pressure. It is known that qE quenching involves converting violaxanthin to zeaxanthin, which is generally associated with desiccation tolerance in lichen [55,56]. Other processes protecting the photosynthetic apparatus during desiccation are the increase in antioxidant content (see [57]), conformational changes of pigment–protein complexes, and thermal dissipation of absorbed light energy [17], dehydration-induced PSII deactivation (see [35]), xanthophyll cycle-independent mechanisms [14], and efficient spillover, i.e., energy transfer from PSII to PSI and consequent quenching due to the formation of a long-lived P700+ state [58]. The PSII to PSI energy transfer is attributed to chlorophyll molecules aggregation in the LHCs of PSII or a new type of quenching in the PSII core antenna [59]. Generally, the protective nonphotochemical dissipation of absorbed light energy happens fast in lichens and other desiccation-tolerant organisms [18,60].

Because nonphotochemical quenching increases with desiccation, the overall ChlF signal decreases. This has been demonstrated for the decline rate of F_0_ and Fs during desiccation at RWCs of 30–40% (see Figure 6 and Figure 7). The desiccation-induced decrease in F_0_ in lichen is attributed to changes in the optical properties of the upper cortex [61] and increased nonradiative dissipation of absorbed excitation energy from light-harvesting complexes [59,62].

### 4.2. Kautsky Curve Changes during Desiccation

Our data on the OPSM shape of KKs is in agreement with a previous study [63], which reviewed the main differences between “algal” and “cyanobacterial” curves. Except for *L. brialmontii* and cephalodia of *P. antarctica* (flat KK), the lichen species exhibited distinguished O, P, S, M, and T points in the wet state (RWC = 100 %), with the ChlF level at P point lower than that at M point on the KK. For higher plants and algae, however, higher ChlF levels reached in the P-S than the M-T part of KKs are reported [20,63] similarly to findings for the *P. antarctica* algal part (fully hydrated, Figure 7). In chlorolichens, such KKs having higher ChlF levels in the P-S than the M-T part of KKs have been observed (e.g., [30]—Conti et al. 2014—*Usnea antarctica*). For some other Antarctic lichens, however, lower P-S than M-T ChlF levels are reported [40,64]. The latter study reports also the light effects on the ChlF levels at P and M points, and, consequently ChlF P/M ratio (see Table 4).

When lower values of ChlF are reached in the P-S than in the M-T part of KKs in cyanobacteria, this is attributed to State 2→State 1 transition taking place during the S to M rise (see e.g., [65,66]). For unicellular green algae, it is attributed to the migration of LHCII from PS I to PS II after LHC dephosphorylation (e.g., [67] for *Chlamydomonas reinhardtii*). In chlorolichens and their symbiotic alga *Trebouxia* sp., however, the mechanism has not been experimentally proven.

In *R. terebrata*, *H. lugubris*, *P. saxatilis*, and *P. antarctica* (algal part), the M peak (see Figure 7) tended to decrease with decreasing RWC, which can be attributed to the involvement of state transitions [63]. As shown in an earlier study [68], the M peak tends to decrease with higher light intensity. The study reported a decrease corresponding to the light increase from 40 to 160 μmol m^−2^ s^−1^ in *Umbilicaria antarctica*. The same study reported a higher ChlF at P than at M point for *Dermatocarpon polyphyllizum* (*c.f. P. antarctica* in Figure 7). Moreover, the P/M ratio of ChlF is temperature-dependent because the KKs shape differs at different temperatures when measured in an optimally hydrated state. The time at which the M point occurs is also temperature-dependent; however, the M point is typically found 60–140 s after the actinic light starts to induce the KKs. In our experiment, the M point was found at varying intervals, ranging from 30 s to 60 s. Except for *L. brialmontii*, the experimental species exhibited similar responses (trends) in O, P, S, M, T, and ratios (Table 4) with thallus desiccation. In general, numeric values of the ratios P/S, P/M, S/M and M/T increased with desiccation, which is consistent with [60] reporting an increase in the parameters with severity of temperature stress and flattening of the KKs curve. Therefore, the differences between ChlF values at the P, S, M, T decreases, and numeric value of the above-specified ratios increases. In some species, however, the parameters did not show any change with desiccation, which can be explained by generally small differences between P, S, M, and T ChlF levels in “flat” KKs recorded during desiccation (e.g., *R. terebrata*, *P. antarctica*—algal part). This contrasts with an earlier study [40] reporting a wide range of numeric values of P/S and P/M in wet Antarctic lichen exposed to a decreasing thallus temperature. In *L. brialmontii*, where the ratios (Table 4) were close to 1, the phenomenon can be explained by flat KKs with small differences between the P, S, M, and T points, which were hardly distinguishable. A flat KKs is more typical for cyanolichens [9] than chlorolichens. However, chlorococcoid photobiont has been reported for *Lecania* species [69]. Therefore, we recommend that follow-up studies should conduct a detailed analysis of the KKs shape and quenching mechanisms.

The KKs provide more details about the species-specific responses because the fruticolose *Himantormia lugubris* in maritime Antarctica grows in a different microenvironment than the crustose *Placopsis antarctica*. While both species grow on rocks and stones, *P.antarctica* presents a higher ChlF signal than *H. lugubris* at 100% RWC (Figure 7), which suggests that *P. antarctica* is photosynthetically more active at water saturation than the latter. In this context, it is important to mention that *H. lugubris* is a species that is rather tolerant to drought and does not tolerate water saturation like other Antarctic lichens which show carbon assimilation decrease with water saturation [13]. On the other side, *P. antartica* is a pioneer species on glacier retreat areas, which are prone to high humidity. Another species that is not as active at high RWC is *L. brialmontii*, as its fluorescence signal in the KKs study shows that ChlF changes insignificantly from 19–100%, with its activity being slightly lower at 100% (Figure 7).

### 4.3. Lichen Spectral Properties

Species-specific spectral curves could be correlated with the color of thalli. No change in the spectral reflectance spectrum in wet and dry thalli of *H. lugubris* can be attributed to the generally black thallus that reflects similarly in wet as well as dry conditions. A similar pattern was found for the blackish cyanolichen *Leptogium puberulum* (reflectance below 0.1) [33,70].

For *L. brialmontii*, the wet-state spectral reflectance curve had lower reflectance values than the dry one. However, for both, a biphasic increase in the red-edge region was found as follows: there was a higher rate within the ranges of 680 nm to 710 nm and 710 nm to 800 nm, respectively. The presence of a red-edge increase in spectral reflectance is a characteristic of chlorolichens that has been reported and documented for many species that exhibit a large variety of thallus colors (see [71,72]). In general, hydrated chlorolichen samples, except for the black *H. lugubris*, showed lower reflectance values than in the dry state. This is mainly true for the spectral range of 400–670 nm, where the difference between the wet and dry spectra is most remarkable (Figure 8), as shown earlier by [64] for Antarctic lichen *Dermatocarpon polyphyllizum.* Hydration-induced reflectance changes in the visible region are due to the high absorption by photosynthetic pigments, while high reflectance in the near-infrared region is due to high scattering from the internal tissues of the lichens [73,74].

The shape of the spectral reflectance curve of *R. terebrata* is typical for lichens with green thalli, with several local maxima in the wavelength range of 520–660 nm followed by the red-edge increase. Similar features of the respective curves were found for the thalli of two Antarctic species which are green when moist but not in dry state: *Physconia muscigena* [33], and *Dermatocarpon polyphyllizum* [75]. The spectral range of 520–660 is affected mainly by the composition and number of photosynthetic pigments which are sensed as the above-mentioned maxima in wet thalli but not in dry thalli which appear grey (*P. muscigena*, *D. polyphillizum*) due to the chromatic change in the dehydrated upper cortex. The role of the upper cortex optical properties can be seen in the spectral curves of the algae-containing part of *Placopsis antarctica* which, due to generally white color in both dry and wet states, shows almost identical spectral curves (see Figure 8) and does not have any local maxima in the 520–660 nm band.

Our data suggest that apart from species-specific effects, NDVI and PRI indices differ in dry and wet lichens. An increase in NDVI in dry vs. wet thalli in black *H. lugubris* can be compared to a similar response in the following polar lichens that have a black thallus in a wet state: *Umbilicaria hirsuta* [76], *U. cylindrica*, and *U. decussata* ([77]). However, other representatives of Umbilicariaceae that have a deep green color in wet state show a decrease of NDVI with thallus desiccation, as well as lower NDVI values in dry than in wet condition (*U. arctica*, *U. hyperborea*: see [49]).

Our result shows that reflectance curves and/or NDVI offer great potential for remote sensing studies, at least for the detection of Antarctic lichen, according to differential physiological conditions. In many lichen species, NDVI differs in wet and dry states. Typically, lower NDVI values are found in a wet than a dry state [33]. Therefore, NDVI values would be used as proxies of hydration status of thalli and physiological activity of lichens. With technological progress and more frequent use of unmanned aerial vehicles for follow-up studies in Antarctic terrestrial ecosystems, an improvement in the application of NDVI in lichen ecophysiological studies might be expected.

## 5. Conclusions

Dehydration response curves of potential yield of PSII (F_V_/F_M_) revealed that primary photosynthetic processes of PSII remained active in *P. antarctica* cephalodia, even after severe desiccation (RWC below 10%) compared to the chlorolichens that exhibit a high degree of F_V_/F_M_ limitation. This might be explained by the effect of the exopolysaccaridic envelope of symbiotic *Nostoc commune* that helps the cephalodium to maintain higher RWC than average values for the whole thallus including the algal part of the thallus.For effective yield of PSII (Φ_PSII_), the dehydration response curves indicated that *H. lugubris* showed somewhat higher Φ_PSII_ values at the RWC < 10% than the rest of the investigated species. Together with gradual activation of nonphotochemical quenching (NPQ) during thallus desiccation, this suggests an advantage for the species in terms of efficient primary photosynthetic processes during the final phase of desiccation. This might be a useful mechanism because in the field, the species desiccates rapidly due to the black thallus color. Generally, lichens with dark thalli absorb more light energy and therefore warm up and desiccate more rapidly than species with brighter-colored thalli.All species showed a decrease in the ChlF signal with ongoing desiccation and general flattening of the slow Kautsky kinetics curve. As a result of this phenomenon the index parameters derived from the ChlF levels P, S, M, S, such as P/S, P/M S/M generally increased, but the response and indicative value of such parameters for ecophysiological studies must be verified in follow-up studies.Spectral reflectance curves recorded in wet and dry states of thalli showed more evident differences in green than black or blackish lichens, with typically lower reflectance in the wet than in the dry state.In addition to being used to separate rock-dwelling lichens from those growing on bare soil, species-specific spectral signatures, spectral reflectance methods, and NDVI in particular have the potential to determine lichen physiological activity and water content [78].

## Figures and Tables

**Figure 1 microorganisms-09-00818-f001:**
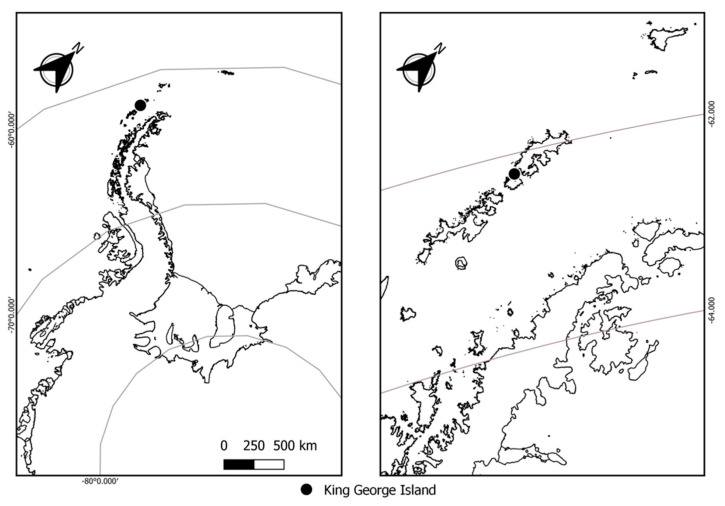
Map of the Antarctic Peninsula showing the northern part of the South Shetlands Archipelago (**left**) and the localization of King George Island (South Shetlands, Antarctica). The point indicates Fildes Peninsula (**right**) according to the data set in https://doi.org/10.5285/ad7d345a-0650-4f44-b7eb-c48e1999086b (accessed on 1 February 2021).

**Figure 2 microorganisms-09-00818-f002:**
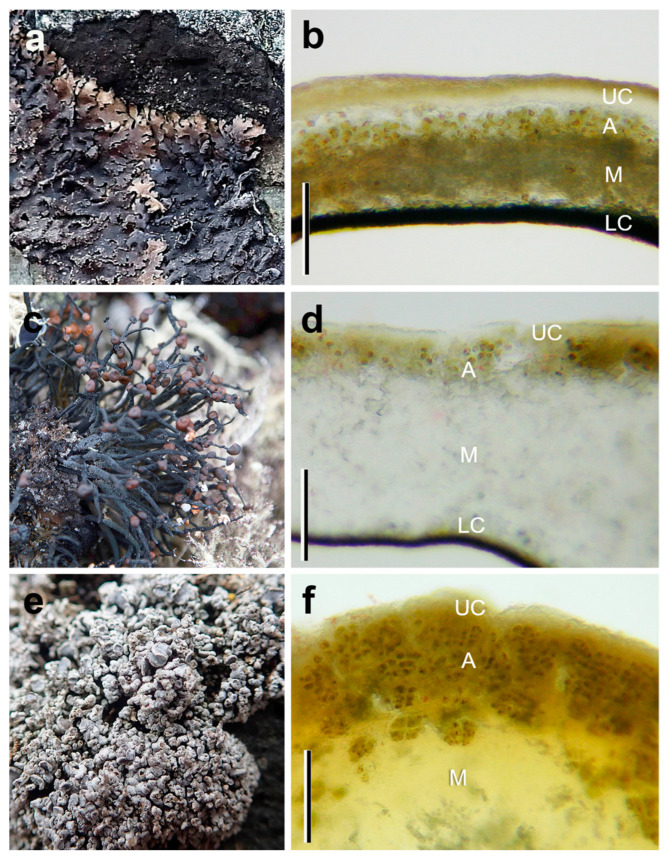
In situ morphology and the anatomical characteristics of Antarctic lichens used in the experiments showing the four thalline layers, upper cortex (UC), green-algae layer (A), medulla (M), and lower cortex (LC). (**a**,**b**) the foliose *Parmelia saxatilis*; (**c**,**d**) a fruticulose specimen of *Himantormia lugubris*; (**e**,**f**): the microfruticolose cushion *Lecania brialmontii* (bar = 100 μm). Photo credit: A. Casanova-Katny.

**Figure 3 microorganisms-09-00818-f003:**
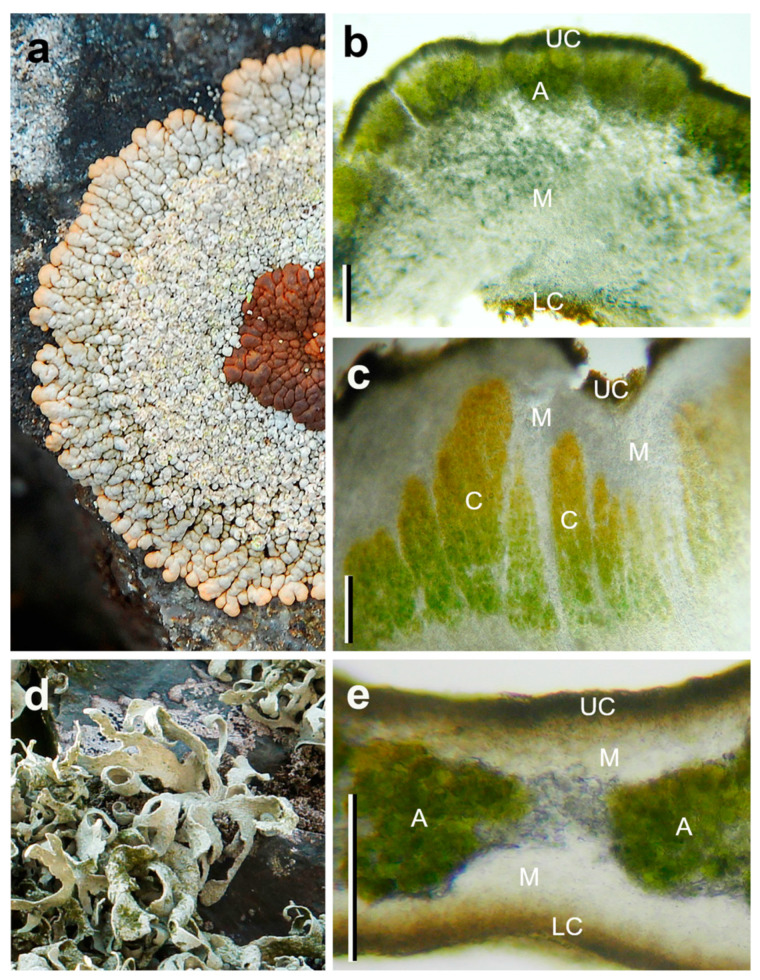
In situ morphology and anatomical characteristics of Antarctic lichens used in the experiments showing the four thalline layers, upper cortex (UC), green-algae layer (A), medulla (M), cephalodium (C), and lower cortex (LC). (**a**) the crustose *Placopsis antarctica* thallus; (**b**) green photobiont section, (**c**) cephalodium section; (**d**,**e**) the fruticolose *Ramalina terebrata* (bar = 100 μm). Photo credit: A. Casanova-Katny and A. Beck (*R. terebrata*).

**Figure 4 microorganisms-09-00818-f004:**
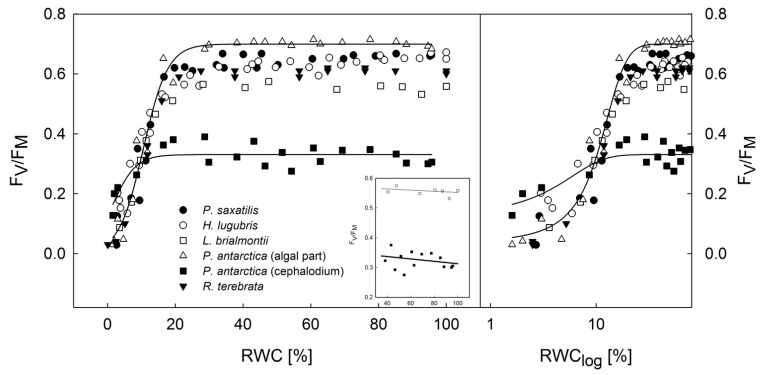
Desiccation response curve of F_V_/F_M_ in experimental lichen species desiccating from a wet (RWC = 100%) to a dry state (RWC = 0%) expressed in the decadic (**left**) and logarithmic RWC (**right**) scales. The fits with distinguishable S-curves at a RWC of below 20% are presented for the algal and cyanobacterial (cephalodium) parts of *Placopsis antarctica.* The inset shows a linear fit through the *H. lugubris* and *P. antractica* (cephalodium) data points within the RWC range of 30–100%.

**Figure 5 microorganisms-09-00818-f005:**
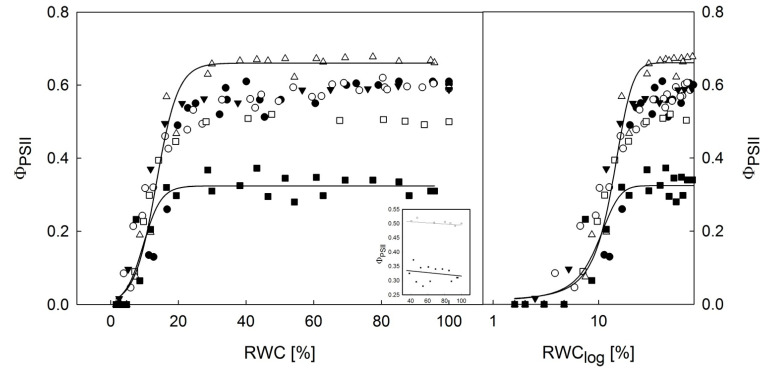
Desiccation response curve of effective quantum yield (Φ_PSII_) in experimental lichen species desiccating from a wet (RWC = 100%) to a dry state (RWC = 0%) expressed in decadic (**left**) and logarithmic RWC (**right**) scales. The fits with distinguishable S-curves at a RWC of below 20% are presented for the algal and cyanobacterial (cephalodium) parts of *Placopsis antarctica.* The inset shows a linear fit through the *H. lugubris* and *P. antractica* (cephalodium) datapoints within the RWC range of 30–100%.

**Figure 6 microorganisms-09-00818-f006:**
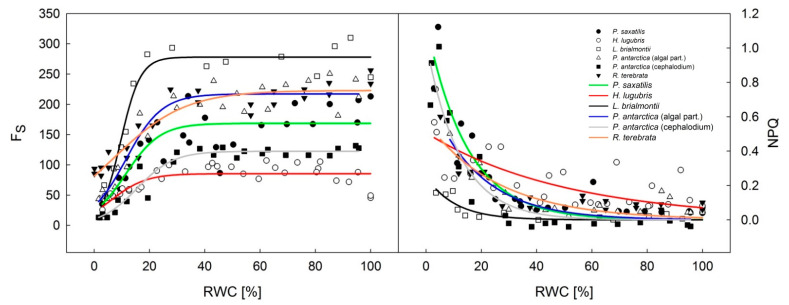
Desiccation response curve of steady-state chlorophyll fluorescence (F_S_—(**left**) panel) and nonphotochemical quenching (NPQ—(**right**) panel) in Antarctic lichens desiccating from a wet (RWC = 100%) to a dry state (RWC = 0%).

**Figure 7 microorganisms-09-00818-f007:**
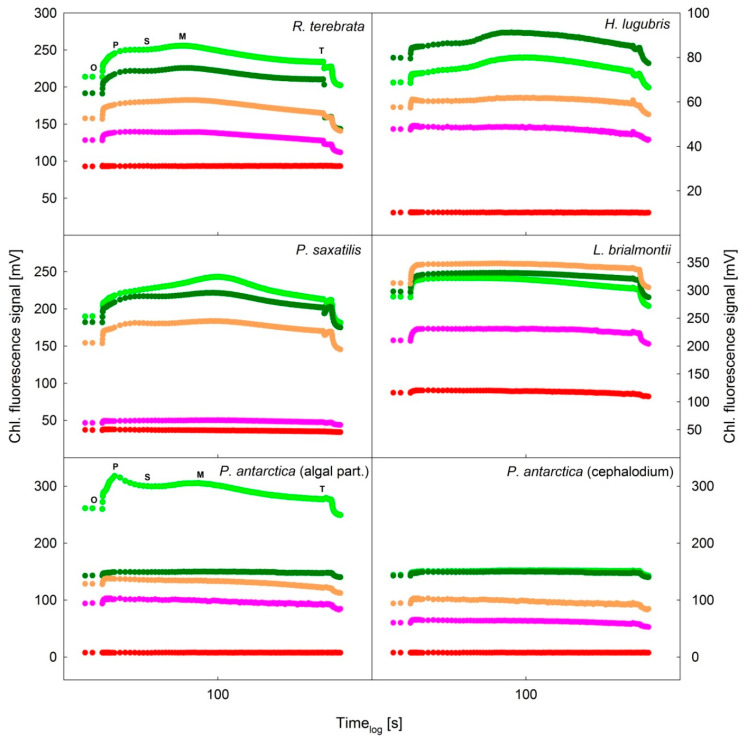
Slow Kautsky kinetics recorded for desiccating lichens at five different RWCs, which varied according to the lichen species. *Ramalina terebrata* (RWC = 100%, 65%, 16%, 5%, 0.1%), *Himantormia lugubris* (RWC = 100%, 65%, 16%, 6%, 3%), *Parmelia saxatilis* (RWC = 100%, 72%, 22%, 9%, 3%), *Lecania brialmontii* (RWC = 100%, 67%, 19%, 9%, 4%), *Placopsis antarctica* (RWC = 100%, 62%, 19%, 5%, 1%), and *P. antarctica* cephalodium (RWC = 100%, 62%, 19%, 5%, 1%). The RWC values are provided in the following different colors: 100% = soft green; 62–72% = deep green; 16–22% = orange; 5–9% = violet; 0.1–4% = red. The time axis (x) is given in a log scale. The O, P, S, M, and T points are indicated for *R. terebrata* and *Placopsis antarctica* (algal part of the thallus). The chlorophyll fluorescence signal is shown with different scales according to the species-specific values.

**Figure 8 microorganisms-09-00818-f008:**
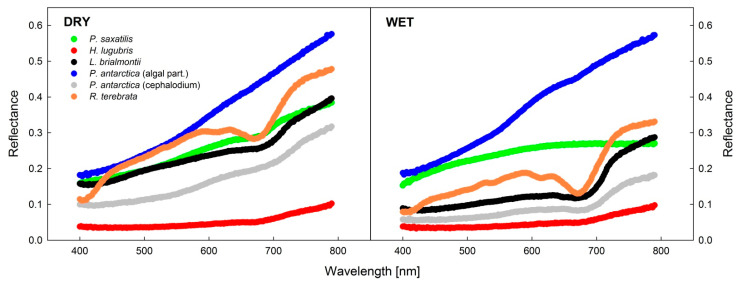
Spectral reflectance curves recorded for experimental lichen species in dry (RWC = 0%) and wet (RWC = 100%) states of the thallus.

**Table 1 microorganisms-09-00818-t001:** Antarctic lichen species used in the study from Fildes Peninsula, King George Island. The species’ details were obtained from field observation and using [1,2,39].

Lichen Family	Lichen Species	Photobiont	Thallus Morphotype	Ecology and Distribution
Parmeliacea	*Parmelia saxatilis* (L.) Ach	Trebouxioid	Foliose	Cosmopolitan species with southern limit in Antarctica, on rocks, boulders, stones, and mosses, and on dry to moist rock faces. Apothecia not observed in Antarctica.
	*Himantormia lugubris* (Hue) I.M. Lamb	Trebouxioid	Fruticose	Endemic in Antarctica. Preferentially saxicolous on acidic rocks and abundant on the southern part of Fildes Peninsula and Ardley Island, where it forms several patches on soil or moss. In nitrophobic communities, growing above 70 m.a.s.l.
Ramalinacea	*Lecania brialmontii* (Vain.) Zahlbr.	Trebouxioid	Fruticose	Endemic in Antarctica. Preferentially saxicolous, grows on rocks and moist and shaded sites, typical for ornithocoprophilous lichen communities.
	*Ramalina terebrata* Hook. F. & Taylor	Trebouxioid	Fruticose	Cosmopolitan species with southern limit in Antarctica. Saxicolous, grows on costal cliffs and large boulders, typical for ornithocoprophilous lichen communities. Apothecia not observed in Antarctica.
Trapeliaceae	*Placopsis antarctica*, D.J. Galloway, R.I.L. Sm. & Quilhot	*Stichococcus antarcticus*; or *S. allas*	Crustose	Endemic in Antarctica. Grows in nitrophobic communities. Cyanobacteria in cephalodia, with species of *Nostoc*.

**Table 2 microorganisms-09-00818-t002:** List of used spectral reflectance indices with equations and referenced sources.

Spectral Reflectance Indices	Equation	References
Normalized Difference Vegetation Index (NDVI)	NDVI = (R_780_ − R_670_)/(R_780_ + R_670_)	[41] Rouse et al. (1974)
Photochemical Reflectance Index (PRI)	PRI = (R_531_ − R_570_)/(R_531_ + R_570_)	[42] Gamon et al. (1992)

**Table 3 microorganisms-09-00818-t003:** Regression coefficient (R^2^) for the nonlinear relationships between Fs (steady-state chlorophyll fluorescence and potential (F_V_/F_M_) and effective quantum yields of PSII (Φ_PSII_) for lichen species. The regression model was a fourth order polynomial.

Lichen Species	R^2^ for Fs Versus	
	F_V_/F_M_	Φ_PSII_
*Parmelia saxatilis*	0.9777	0.9804
*Himantormia lugubris*	0.5265	0.3106
*Lecania brialmontii*	0.9768	0.9752
*Placopsis antarctica*—algal part	0.9966	0.9954
*Placopsis antarctica*—cephalodium	0.4766	0.8825
*Ramalina terebrata*	0.9098	0.9211

**Table 4 microorganisms-09-00818-t004:** Parameters derived from the O, P, S, M, T curves (means of five replicates per species and particular RWC). The values were distinguished on the slow Kautsky kinetics recorded for the experimental lichen species during their desiccation from a wet (RWC = 100%) to a dry state (RWC below 5%).

Lichen Species	RWC (%)	Parameters					
		P/S	P/M	S/M	M/T	O/P	O/T
*Lecania brialmontii*	100	0.990	0.991	1.000	1.058	0.905	0.949
	67	0.987	0.984	0.997	1.032	0.910	0.925
	19	0.996	1.006	1.010	1.011	0.902	0.918
	9	1.000	0.998	0.998	1.037	0.907	0.939
	4	1.004	1.003	0.999	1.051	0.962	1.014
*Himantormia lugubris*	100	0.973	0.911	0.936	1.081	0.942	0.927
	65	0.984	0.930	0.945	1.071	0.937	0.933
	16	1.009	0.982	0.974	1.045	0.942	0.968
	6	1.015	1.006	0.992	1.078	0.966	1.047
	3	1.004	1.007	1.003	1.002	0.995	1.004
*P. antarctica* algal part	100	1.060	1.042	0.983	1.101	0.822	0.943
	62	1.000	0.993	0.993	1.018	0.957	0.968
	19	1.015	1.022	1.007	1.105	0.931	1.051
	5	1.017	1.012	0.995	1.110	0.910	1.022
	1	1.003	0.999	0.996	1.024	0.987	1.009
*P. antarctica* cephalodium	100	1.008	0.991	0.983	1.005	0.950	0.947
	62	1.001	0.995	0.995	1.017	0.958	0.969
	19	1.009	1.004	0.996	1.085	0.938	1.022
	5	1.016	1.031	1.015	1.087	0.915	1.025
	1	1.027	1.010	0.984	1.001	0.932	0.943
*Ramalina terebrata*	100	0.973	0.953	0.979	1.092	0.855	0.890
	65	0.982	0.963	0.981	1.074	0.879	0.909
	16	0.970	0.956	0.986	1.097	0.899	0.944
	5	0.987	0.989	1.002	1.090	0.930	1.002
	0.5	0.999	0.998	0.999	1.000	0.995	0.993
*Parmelia saxatilis*	100	0.963	0.898	0.932	1.136	0.871	0.888
	72	0.963	0.943	0.979	1.096	0.870	0.899
	22	0.970	0.953	0.983	1.075	0.878	0.900
	9	1.007	0.981	0.974	1.058	0.941	0.976
	3	1.010	1.043	1.033	1.029	0.972	1.043

**Table 5 microorganisms-09-00818-t005:** Means of spectral indices (NDVI, PRI) recorded for dry and wet thalli of the five Antarctic lichen species (means of five replicates per species in wet and dry state). Different upper index letters indicate statistically significant differences at *p* ≤ 0.05.

Spectral Index	State of the Thallus	Lichens Species					
		*Himantormia lugubris*	*Lecania brialmontii*	*Ramalina terebrata*	*Parmelia saxatilis*	*Placopsis antarctica* (green algae)	*Placopsis antarctica* cephalodium
NDVI	Dry	0.329 ^a^	0.213 ^a^	0.212	0.152 ^a^	0.033 ^a^	0.190 ^a^
	Wet	0.23 ^b^	0.39 ^b^	0.300 ^b^	0.060 ^b^	0.143 ^b^	0.211 ^a^
PRI	Dry	−0.059 ^a^	−0.038 ^a^	−0.058 ^a^	−0.061 ^a^	0.003 ^a^	−0.056 ^a^
	Wet	−0.029 ^b^	−0.039 ^a^	−0.054 ^a^	−0.075 ^a^	−0.082 ^b^	−0.077 ^b^

## Data Availability

Not applicable.

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
