# Peer review of "Inhibition of Primary Photosynthesis in Desiccating Antarctic Lichens Differing in Their Photobionts, Thallus Morphology, and Spectral Properties"

_microorganisms, 2021, doi:10.3390/microorganisms9040818_

Round 1

Reviewer 1 Report

The paper by Angelica Casanova-Katny et al. describes the relationship between  water content in selected species of Antarctic lichens and activity of the photosynthetic apparatus.  Species occupying different microbiota and having different photobionts were chosen for the experiments and several photosynthetic parameters were measured.   The aim of this study is interesting and scientifically sound and obtained results are valuable. I consider it as the first and necessary step  for further studies allowing a  deeper understanding how, on a molecular level,  desiccation or rehydration affects various metabolic processes.  A link between these processes and occurence of different water pools in the cells at various steps of dehydration/rehydration and molecular dynamics of water molecules in these pools would be of a special interest.

   I recommend publication of this paper after revision according to remarks given below.

Remarks:

1.      In the list of symbols and abbreviations the following abbreviations are missing:

SRC, Fs, KK, UAV.

2.      Improvement of English is definitely necessary. There are numerous misprints, small errors, in some places style should be improved, some sentences are too long (e.g. second sentence in the Introduction), in many places sentences require re-wording, and in some places the meaning of sentences is not clear.

3.       My concern is about RWC determination, especially the Dw parameter which, according to the Authors, is the weight of the fully dry sample, obtained after 24 h at 35oC. I have serious doubts if the samples are really fully dry after such treatment.  Usually, to obtain dry weight the plant samples are dried for 24 h at 105oC.  It would be interesting to determine how much water still remained in the lichens dried for 24 h at 35oC.

4.      Lines 236 and 239 –  I guess that instead of „thallus size” it should be rather „thallus thickness”.

5.      In description to Fig. 2 should be „e-f” instead of „d-e”.

6.      Fig. 2. In the line 249 „three” should be replaced with „four”.

7.      Fig. 3.   Symbol „C” present on the photos is not explained in the description of this figure.

8.      Line 261. Fig. 3 should be changed to Fig. 4 and Fig. 4 to Fig. 5.

9.      Line 331. „Table 3” should be changed to „Table 5”.

10.   Line 332. „Fig. 3” should be changed to „Fig. 8”.

11.   Table 4. How the O, P, S, M, T points used to calculate P/S, P/M, S/M, M/T, O/P, and O /T parameters were measured.  In Fig. 7 in many cases at lower RWC values there are just straight, flat  lines without any peaks, so positions and values of the mentioned above points are rather not possible to identify and measure.

12.   Lines 383-385.  This seems to be a statement taken from literature so the source paper should be cited.

13.   Line 402. „PSII pressure” should be changed to „PSII excitation pressure”.

14.   Line 421. There is no „M peak” on the Fig. 8. Also changes related to different light intensities are not shown in this Figure.  Perhaps Authors refer to Fig. 8 in the reference [60].

15.   Line 422. The same, as above, concerns Fig. 7.

16.   Conclusions at the end of the Discussion section (a few sentences) would be desirable.I

Author Response

Letter to the reviewers

Dear reviewers, please find below answers to your commnets and suggestions you mentioned in your reviews.

The revised manuscript is supplied in ´all changes accepted´ version (including language revision) and the line references fits to the version:

The other versions supplied are:

  • Manuscript with specific changes indicated (before language correction) and
  • Manuscript with indication of language correction

Reviewer No. 1

Suggestion 1 Some changes into the list of symbols and abbreviations are needed

List of symbols and abbreviation has been revised and addition were done. Lines 38-46

Suggestion 2 Improvement of English is needed

The text of the MS has passed through a language correction procedure. It was done by professional agency. The certificate from the agency could be supplied upon request (http://kontroluje.me).

Suggestion 3 Concerns about dry state of lichens after 24 h in 35 oC

We left the text as it was. In the experiments, we let the lichen thalli to dry during the ChlF measurements which meant they had RWC below 3 % (almost dry). Then we dried them for 24 h in 35 oC and used their weight as reference for dry state (and mentioned this in the text of the MS). The, we put the same thalli into the oven and exposed the for another 24 h to 90 oC and, after that, wigted again. Since the dry weight was the same as prevoius one, i.e. after 24 h in 35oC we considered it as a reference to dry state weight.

Suggestion 4 Change ´thallus size´ to ´thallus weight´

Done

Suggestion 5 Change description of Fig. 2

Done

Suggestion 6 Replace 3 by 4, 4 by 5

Done

Suggestion 7 Explain symbol ´C´

Done

Suggestion 8 Renumber Figs

Fig. 3 was changed to Fig. 4 and Fig. 4 was changed to Fig.5

Suggestion 9 Renumber Table

Table 3 renumbered to Table 5

Suggestion 10 Renumber Figs

Fig.3 was changed to Fig. 8

Suggestion 11 Elucidate how O,P,S,M,T  ChlF levels were determined

In MandM, We implementesd some explanatory lines into MaM describind the method how the O,P,S,M,T points were identified  (lines 209-211) and ChlF levels at these points evaluated. In the measuring software (FluorCam)   or Excel file i tis a routine operation. In Discussion, we added  lines 416-424, 444-447, 518-523.

Suggestion 12 Reference to be added

Yes, we added the citation of original paper and added it into References.

Suggestion 13 rephrase

We rephrased to:  ´PSII excitation pressure´

Suggestion 14 Misleading reference to ´peak M´

Yes, you are right. We refer to Fig. 7 and the sentence was rephrased.

Suggestion 15 Misleading reference

No, it fits. The reference is to Fig.7

Suggestion 16 Conclusions should be added

Conclusions were added

March 5th, 2021                                                                                              The authors

Reviewer 2 Report

I reviewed the manuscript by Bartak et al. entitled “Inhibition of primary photosynthesis in desiccating Antarctic lichens differing in their photobionts, thallus morphology, and spectral properties”. Difference of photosynthetic properties in five Antarctic lichen species seems interesting, however, I have some serious concerns on this manuscript as follows:

  1. The middle part of the introduction is almost the same to that in the published article (Mareckova and Bartak, 2016 Czech Polar Rep). Although this manuscript is strongly linked to the previous one, the level of similarity is inappropriate in my opinion.
  2. The concept of this study is strongly overlapping with a pervious study, Bartak et al., 2018. Difference of key photosynthetic parameters in Antarctic lichens under desiccation stress were already addressed in the 2018 study. Investigated lichen species and habitats were different in this manuscript, but the method and conclusion were not essentially updated. Since the manuscript in this journal should give “a substantial amount of new information”, I think the framework of this work must be fundamentally upgraded.
  3. The conclusion of this work is too ambiguous. I would suppose that the data should be discussed from the physiological and ecological perspectives. The authors investigated a number of photosynthetic parameters in lichens, but they did not describe the meanings of those values. Particularly, the physiological background of the ratio represented in the Table 4 was not explained at all. Unless such discussion, this work cannot contribute to understand the ecology of lichens.

Every phases of the Kautsky kinetics correspond to particular photosynthetic process as reviewed in Papageorigiou et al. (2007, Photosynth Res), for example. Moreover, induction pattern should be interpreted in different ways in algae and cyanobacteria. I would recommend the authors to reconsider these points more carefully and discuss which process of photosynthesis showed interesting response to the drought stress in each species.

  1. The discussion is highly speculative and confusing. For instance, the authors declare that the Antarctic lichens are tolerant to desiccation (l. 362), however, the evidence for this statement is not clearly described. Comparison with other species is required for such argument. They also mentioned that P. antarctica and L. brialmontii showed different Fv/Fm and Φ(PSII) under drying conditions compared to other species (Ll. 366-367), however, I could not find any distinguishable profiles in L. brialmontii. Moreover, it is too difficult to understand why the authors speculate that qE is the major component in NPQ of lichen under desiccating conditions (Ll 402-409). In ll. 310-312 and ll. 450-452, the authors indicate the effect of morphology and spectral properties on photosynthetic properties in lichens, but the actual relationship is not clearly described and even partially contradict with the context in the final section. For these reasons, I failed to capture the conclusion of this manuscript.
  2. The description of the method for the Kautsky kinetics must be improved. Since some of the induction curves are almost flat, it is necessary to describe how the P, S, M and T positions were identified. Without such description, it is not possible to qualify the data in Table 4. I assume that “Fm” in l. 312 is different from Fm for Fv/Fm, but there are no explanation on it. In addition, the authors should show how many biological replicates were tested and represent the statistical values particularly for Tables 4 and 5.
  3. Data in Figure 6 (NPQ) and Table 4 must be explained in the result section.
  4. Extensive language editing is needed to make this manuscript more straightforward.

Overall, I think this work is highly preliminary and would like to encourage the authors to make an extensive effort to improve the significance of this manuscript.

Author Response

Letter to the reviewers

Dear reviewers, please find below answers to your commnets and suggestions you mentioned in your reviews.

The revised manuscript is supplied in ´all changes accepted´ version (including language revision) and the line references fits to the version:

The other versions supplied are:

  • Manuscript with specific changes indicated (before language correction) and
  • Manuscript with indication of language correction

Reviewer No. 2

The team of authors would like to express thanks for careful reading of the text by the reviewer 2 which resulted in numerous suggestions that undoubtedly led to the improvements of the manuscript. In the following text we explain the changes.

General changes

The text of the MS has passed through a language correction procedure. It was done by professional agency. We are enclosing the certificate from the agency.

Severeal new references has been added in order to support specific parts of Introduction  (lines 87-135) and Discussion.

Conclusions have been added.

Suggestion 1 (To change the middle part of Introduction)

It was completely rewritten and rephrased with several new citations added.  Lines 85-114.

Suggestion 2 (The concept is similar with the previous study. Suggestion: to change)

Yes, you are right. General concept is similar since the same method was used i.e. evaluation of chlorophyll fluorescence parameters (by ChlF imaging technique) during thallus desiccation from RWC of 100% to the RWCs close to 0%. This approach is generally used in the Brno lab. However, in the MICROORGANISMS manuscript, there were several addition such as e.g. spectral reflectance curves and more detailed analysis of the Kautsky kinetics curve shape (OPSMT). In the revised version of MS, we added several new paragraphs into the discussion, mainly related to OPMST the index parameters derived from OPSMT and spectral reflectance.

Suggestion 3 (physiogical background of the OPSMT and the OPSMT parameters presented in Table 4 should be added)

These aspects have been described in Discussion – newly added text related to the underlaying mechanisms. Lines 416-424, 444-447, 518-523.

Suggestion 4 (Discussion should be improved in the specific parts)

Tolerance of Antarctic lichens to desiccaton

                It was repharased (lines 142-144)

Energy-dependent quenching is major component of NPQ.

This was too brief and missleading statement. We put the reference to the paper considering qE role in NPQ in lichens and rephrased the statement (lines 419-421).

The effects of spectral properties and thallus morphology on photosynthetic parameters.

You are right. This was rather general statement having not much support in presented data. The effects are highly species-specific related e.g. to a 3-D structure of thallus that (especially in overlaping and closely packed thallus parts) may lead to limitation of CO2 intake and highly different amount(s) of absorbed light energy. However, these aspects have not been investigated in the study. Therefore, we rejected the statement from the text.

Suggestion 5 The description of Kautsky kinetics analysis must be improved.

In M&M, We implemented some explanatory lines into MaM (lines 209-211) describing the method how the O,P,S,M,T points were identified and ChlF levels at these points evaluated. In the measuring software (FluorCam) or Excel file, it is a routine operation. It is, however, difficult to refer to a certain time, at which the particular ChlF leves are reached. The time(s) differed not only between species but also in between the samples taken from a particular species.

Suggestion 6 Data in Fig. 6 should be explained in more details

We added few new lines into Results (lines 304-306), and Discussion (lines 412-414)

March 5th, 2021                                                                                              The authors

Round 2

Reviewer 2 Report

I reviewed the revised version of the manuscript by Bartak et al. The authors made considerable improvement on their manuscript, however, I still have a number of concerns mainly on the newly added part.

Major points

  1. I think further language editing is needed on the part newly added in this version. Some sentences in Discussion and Conclusion are not easy to understand precisely and severely impairing the impact of this manuscript.
  2. Although the precise description about RWC1/2 is very helpful, I still failed to recognize how the authors evaluate the drought tolerance of each species. In Ll 388-91, P. saxatilis and H. lugubris are suggested to have highest tolerance but the RWC1/2 value of these species did not show remarkable difference from others.
  3. Ll 470-478, I failed to understand how this paragraph contributes to understand the phenotype of Antarctic lichen species.
  4. I think the fourth point of the conclusion is misleading. As the authors mentioned in their response, the relationship between the spectral features and physiological activity is not explained in this manuscript. Please reconsider the way of writing.
  5. Authors should show how many biological replicates were tested and represent the statistical values particularly for Tables 4 and 5. This point was not solved in this round.

Minor points

L 27, 22-32%. These exact numbers are not found in the main text. Please make them consistent.

L 98, P, S and M peaks --> P-S-M transition. Is it correct?

Ll 103, 395, 404, 464, Remove “by” before the number of citation and add some words if needed. e.g. was reported by [xx] --> was reported previously [xx]

Ll 117, 121, Please explain what Fv/Fm, ΦPSII and Fs indicate.

Ll 127-8, Please describe more precisely about “basic physiological response of primary photosynthetic processes”.

L 128, In “this” study?

L 194, Could the authors provide actual intensity (µmol photons m-2 s-1) of actinic light?

L 265, Was P. antarctica only species that showed higher Fv/Fm and ΦPSII under 30% RWC than 100%? Or was it also observed in L. brialmontii? The latter seems to be the case according to the insets of Fig 4 and 5.

Ll 266-7, Does this sentence mean that Fv/Fm and ΦPSII started to decrease at RWC 30% and 20%, respectively? Please rephrase this part since it is not clear enough.

L 271, higher RWC1/2 values?

L 273, What does “significantly low values” indicate? Did the authors compare RWCcrit in several species?

L 276, I failed to understand the last phrase, “as significant points were found at similar RWCs”. Please consider to rephrase it.

L 305, The two curves showed less increase --> The two species showed smaller NPQ increases

L 323, Please describe the details of species specificity.

L 324, Which species were included in “they”? Please explain Fp and Fm as well.

L 381-2, I would suggest to describe the difference more in detail.

Ll 394-9, This paragraph is redundant with the previous one. Please rethink of the flow of discussion.

Ll 416-7, This sentence is not well connected to others in my opinion.

L 426, Please describe “latter” more precisely.

Ll 429-32, Please explain how the authors think about the relationship between the optical properties and NPQ in lichen species they studied.

L 436, with [63], who --> with a previous study [63], which

L 438, I think that the algal part of P. antarctica is also the exception. Please confirm.

Ll 439, 444, I think it is difficult to understand “higher” or “lower” P-S level, because P-S is likely to indicate the transition of chlorophyll fluorescence. Please reconsider about these phrase.

Ll 441-3, Could the authors explain the reason of the different fluorescence profile in each study?

L 453, ted to --> tend to

L 454, higher ChF --> higher ChlF

Ll 460-1, I failed to understand this sentence. Could the authors rephase it?

Ll 472-4, I think activation of carbon fixation decrease the chlorophyll fluorescence in general. Please confirm.

Ll 497-8, A similar response was would in ... below 0.1) [33, 71].

L 541, i”n”cluding

L 546, considered “as an” useful mechanism

L 547, Why the black color cause rapid desiccation?

L 556, Delete “e.g.”; promi”s”ing.

Ll 560-2, Does “signatures” mean “features”?

I hope my comments contribute to improve the significance of this manuscript.

Author Response

LETTER TO THE REVIEWER

Dear reviewer,

Thank you once again for careful reading and the suggestions you mentioned in your review. We have changed the manuscript according to your suggestion. In the below list, we describe the changes done in the manuscript after the second revision (R2 version, March 18th, 2021.) that we sent to the MICROORGANISMS Editors.

Major changes

  1. I think further language editing is needed on the part newly added in this version. Some sentences in Discussion and Conclusion are not easy to understand precisely and severely impairing the impact of this manuscript.

English of the newly added parts of the text as well as whole manuscript (R2) have been cheked by Dr. Goetz Palfner – fluent English speaker who spent several years in UK. The language-related changed are indicated in the R2 version of the MS by tracking changes option (´GFPalfner´).

Prevoius (R1) version of the MS was revised by a professional language company 2 weeks ag.

  1. Although the precise description about RWC1/2 is very helpful, I still failed to recognize how the authors evaluate the drought tolerance of each species. In Ll 388-91, P. saxatilis and H. lugubris are suggested to have highest tolerance but the RWC1/2 value of these species did not show remarkable difference from others.

Statements about drought tolerance were enlarged. Information about species-specific numeric values of critical RWC for Fv/Fm, and FPSII were added into Results. (lines 314-332)

  1. Ll 470-478, I failed to understand how this paragraph contributes to understand the phenotype of Antarctic lichen species.

You are right. The paragraph described rather the factors affecting the slow Kautsky kinetics (OPSMT) than interspecific differences in the five studied lichens. Therefore, we rejected it from the R2 manuscript.

  1. I think the fourth point of the conclusion is misleading. As the authors mentioned in their response, the relationship between the spectral features and physiological activity is not explained in this manuscript. Please reconsider the way of writing.

The fouth point of Conclusions was reduced by 80%. Remaining 20% statement refers to the differences in spectral reflectance curves recorded in wet and dry state of the 5 experimental species, spectral properties of which have not yet been investigated.

  1. Authors should show how many biological replicates were tested and represent the statistical values particularly for Tables 4 and 5. This point was not solved in this round.

We added the information that 5 replicates were tested. The information is in Table legends (Table 4 and 5) in MS R2.

Minor points

All your suggestion related to formal problems have been implemented into the R2 text and indicated by tracking changes option (see the R2 manuscript with ´changes indicated´).

For majority of your suggestions, we added brief answers (see below).

L 27, 22-32%. These exact numbers are not found in the main text. Please make them consistent.

            We added this information (from Abstract) into the main text as well.

L 98, P, S and M peaks --> P-S-M transition. Is it correct?

            We changed to: The P-S-M  transition.

Ll 103, 395, 404, 464, Remove “by” before the number of citation and add some words if needed. e.g. was reported by [xx] --> was reported previously [xx]

Ll 117, 121, Please explain what Fv/Fm, ΦPSII and Fs indicate.

Since i tis the first mention about the parameters, we added ´full name´of the parameters before their abbreviations. Lines 119-124.

Ll 127-8, Please describe more precisely about “basic physiological response of primary photosynthetic processes”.

We added some explanatory lines and specified the parameters (FPSII, non-photochemical chuencing of absorbed light energy: qN) that were measures in the previous study,. i.e. Acta Physiologiae Plantarum 2018, 40, 177-187. No. 33 in References..

L 128, In “this” study?

Yes. We reprased so that it is easy to understand that the statement relates to the MICROORGANISMS manuscript.

L 194, Could the authors provide actual intensity (µmol photons m-2 s-1) of actinic light?

L 265, Was P. antarctica only species that showed higher Fv/Fm and ΦPSII under 30% RWC than 100%? Or was it also observed in L. brialmontii? The latter seems to be the case according to the insets of Fig 4 and 5.

            Yes, you are right. We added information on Lecania brialmontii.

Ll 266-7, Does this sentence mean that Fv/Fm and ΦPSII started to decrease at RWC 30% and 20%, respectively? Please rephrase this part since it is not clear enough.

            Rephrased.

L 271, higher RWC1/2 values?

            We rephrased.

L 273, What does “significantly low values” indicate? Did the authors compare RWCcrit in several species?

            We added several lines on RWCcrit values for Fv/Fm and FPSII for all species.

L 276, I failed to understand the last phrase, “as significant points were found at similar RWCs”. Please consider to rephrase it.

            It was rephrased and ´significant´rejected.

L 305, The two curves showed less increase --> The two species showed smaller NPQ increases

            Done.

L 323, Please describe the details of species specificity.

            We added two items of specifity.

L 324, Which species were included in “they”? Please explain Fp and Fm as well.

            Rephrased and ´they´ rejected.

L 381-2, I would suggest to describe the difference more in detail.

You are right. The difference does not describe the Fv/Fm and F PSII courses in the two lichen species at the RWCs below 30%. It was ment as the differences in in-situ rapidity of thallus dessication thanks to different microrelief and water avilability. In R2 version, we made it clear. It is ecological but not physiological comment now.

Ll 394-9, This paragraph is redundant with the previous one. Please rethink of the flow of discussion.

            We rejected the paragraph.

Ll 416-7, This sentence is not well connected to others in my opinion.

            We rejected the sentence.

L 426, Please describe “latter” more precisely.

You are right. This was not very clear statement. We changed to. ´ The PSII to PSI energy transfer….´

Ll 429-32, Please explain how the authors think about the relationship between the optical properties and NPQ in lichen species they studied.

We rephrased a bit the statement which relates exclusively to Fo changes (decrease) with desiccation and the changes of the upper cortex optical properties (increased reflectance).

Non-photochemical quenching increases typicall in the RWCs below 30% which is accompanied by activation of protective mechanisms on chloroplastic level. We added reference.

L 436, with [63], who --> with a previous study [63], which

            done

L 438, I think that the algal part of P. antarctica is also the exception. Please confirm.

  1. antarctica (algal part) is not the exception. The statement refers to 100% RWC at which the species shows distiguished P,S,M,T ChlF levels. Chlorophyll fluorescence at point P is higher then fluorescence at M point (at 100 % RWC). It differs from Lecania brialmontii.

Ll 439, 444, I think it is difficult to understand “higher” or “lower” P-S level, because P-S is likely to indicate the transition of chlorophyll fluorescence. Please reconsider about these phrase.

            We rephrased.

Ll 441-3, Could the authors explain the reason of the different fluorescence profile in each study?

According to our yearly experience with KKs in lichen, we have no universal explanation why some chlorolichens exhibit KKs with higher ChlF at P than M point while some do not. However, both KKs shapes have been measured and reported:

            https://journals.muni.cz/CPR/article/view/12814/11130 (Fp higher then Fm)

            https://www.sci.muni.cz/CPR/11cislo/Bartak_Mareckova-web.pdf (Fp lower than Fm)

L 453, ted to --> tend to

                        done

L 454, higher ChF --> higher ChlF

                        done

Ll 460-1, I failed to understand this sentence. Could the authors rephase it?

                        rephrased

Ll 472-4, I think activation of carbon fixation decrease the chlorophyll fluorescence in general. Please confirm.

We rejected whole redundant paragraph including the problematic sentence. Of course, CO2 fixation in Calvin-Benson cycle consumes APT and NADPH which decreases chlorophyll fluorescence throughout whole KK. Therefore, local maxima are not consequence of stimulated dark reaction of photosynthesis (CO2 fxation).

Ll 497-8, A similar response was would in ... below 0.1) [33, 71].

            We rephrased.

L 541, i”n”cluding

                        The typewriting mistake was corrected

L 546, considered “as an” useful mechanism

                        done

L 547, Why the black color cause rapid desiccation?

                        We put several explanatory lines.

L 556, Delete “e.g.”; promi”s”ing.

                        Done.

Ll 560-2, Does “signatures” mean “features”?

No. It means signatures. The term is frequent in remote sensing community and reflects the fact that each surface (including lichen- or moss-dominated vegetation cover) has unique and, in cases of vegetation components, species-specific spectral reflectance curves.

Anyway, in the R2 version of the MS, we rejected the paragraph containing the ´signatures´.

 I hope my comments contribute to improve the significance of this manuscript.

Thanks again for careful reading and suggestions. They undoubtedly helped us to improve the text of the manuscript.

The authors                                                                                         March 18th, 2021

Round 3

Reviewer 2 Report

I reviewed the third version of the manuscript by Bartak et al. The authors solved most of my concerns, but some are remained. In addition, I think some of the interpretations of data are still inappropriate. I would recommend authors to reconsider following points seriously to improve their work.

Major points

Ll 332-3, Didn’t the drop of ΦPSII happen at higher RWC in cyanobacterial parts than algal parts? At least, RWC1/2 is higher in algal parts. In my understanding, the cyanobacterial part can keep its photosynthetic activity at lower RWC than the algal part in return for lower photosynthetic activity under higher RWC, although this is pure speculation. Please reconsider this point carefully.

L 464-5, H. lugubris did not showed higher Fv/Fm, at least, I think. Difference of ΦPSII is also marginal. Please reconsider of this argument.

Ll 601-2, I think high chlorophyll fluorescence does not simply indicate high photosynthetic activity, since if the electron transfer is efficient, fluorescence from chlorophyll decreases. Parameters of photosynthetic activity, Fv/Fm and ΦPSII are higher in H. lugubris than in the cyanobacterial part of P. antarctica. I am very sorry to fail to notice this important point in previous rounds, but please rethink of their relationship.

Minor points

L 392, Please explain what Fp and Fm stand for. I have pointed this out twice, but it has not been answered yet. I think Fm here is completely different from Fm (maximum chlorophyll fluorescence) of Fv/Fm. It would be highly confusing using Fm for both of them without detailed explanation.

L 459, It is a bit confusing which species is “this” lichen here. How about replacing “this” with “L. brialmontii” here?

L 476-7, Since the ROS production in the lichen thallus is speculation whereas the effect of ROS on PSII is generally observed phenomena, I propose to change these sentences as follows:

During..... thallus, excess ROS formation may occur in PSII, ..... High levels of ROS is harmful not only to PSII....

Ll 612-3, What I suggested was to change this sentence to:

“A similar response was found in blackish cyanolichen Leptogium puberulum with similar values (reflectance below 0.1) [33, 71].”

Is this correct? I would apologize if my previous suggestion was confusing.

Author Response

LETTER  TO  THE REVIEWER

April 2nd, 2021

Dear reviewer,

Thank you for the suggestions and comments to the manusrcipt no. Microorganisms-1114555 you mentioned in your review.

In the R3 version, we have accepted all your suggestions and changed the MS accordingly. Please find below list of the changes (they are indicated in R3 MS by tracking changes option as well).

With regards,

Josef Hájek

 //////////////////////////////////////////////////////////////////////////////////////////

Ll 332-3, Didn’t the drop of ΦPSII happen at higher RWC in cyanobacterial parts than algal parts? At least, RWC1/2 is higher in algal parts. In my understanding, the cyanobacterial part can keep its photosynthetic activity at lower RWC than the algal part in return for lower photosynthetic activity under higher RWC, although this is pure speculation. Please reconsider this point carefully.

We rephrased the sentence so that it is clear that the difference between algal and cyanobacterial part relates to the RWC interval from 100 (wet) to 15%. Moreover, we stated that FPSII decline at the RWC below 10% was identical for algal and cyanobacterial part, contrastingly to Fv/Fm where cyanobacterail part retains higher Fv/Fm values in cephalodium that algal part of P. antarctica thallus. Reason for such different ´behaviour´ of FPSII and Fv/Fm at the RWC below 10% remains unclear to us. We would not like to speculate what the reason for this could be without having some supplementary measuremets next year. To postulate a hypothesis we will collect more Placopsis antarctica samples during the next antarctic expedition, which allow us to measure desiccation-response curves of separated cephalodium and algal part. Such approach may give us some insight into primary photochemistry limitations in cephalodium and the algal part at the same RWC. In recent study, the RWC relates to whole thallus which makes it difficult to evaluate differences in dehydration status of cephalodium per se (compared to algal part).                Ref: lines 279-282

L 464-5, H. lugubris did not showed higher Fv/Fm, at least, I think. Difference of ΦPSII is also marginal. Please reconsider of this argument.

We rephrased. Briefly, we rejected Fv/Fm because in this parameter, Himantormia lugubris was not something special in comparison to the other species. To support the idea about high desiccation tolerance in H. lugubris, we added the statement about the lowest NPQ in this species reached at the RWC below 10%.                                 Ref: lines 393-396

Ll 601-2, I think high chlorophyll fluorescence does not simply indicate high photosynthetic activity, since if the electron transfer is efficient, fluorescence from chlorophyll decreases. Parameters of photosynthetic activity, Fv/Fm and ΦPSII are higher in H. lugubris than in the cyanobacterial part of P. antarctica. I am very sorry to fail to notice this important point in previous rounds, but please rethink of their relationship.

                        We rephrased.

L 392, Please explain what Fp and Fm stand for. I have pointed this out twice, but it has not been answered yet. I think Fm here is completely different from Fm (maximum chlorophyll fluorescence) of Fv/Fm. It would be highly confusing using Fm for both of them without detailed explanation.

Yes, you are right. The chlorophyll fluorescence signal (Fm) is attributed to maximum fluorescence reached after the application of saturation pulse (SP) on dark-adapted sample in vast majority of ChlF studies. In the paragraph you have in mind, Fm denotes chlorophyll fluorescence reached at the M point of the OPMST transient (slow Kautsky curve) induced by actinic light, not saturation pulse.

In response to your suggestion, we rephrased the statement and attributed Fm to the M point of the OPSMT. Moreover we explicitly stated that it does not relate to maximum ChlF reached after SP.

Ref: lines 324-328 in R3 manuscript

L 459, It is a bit confusing which species is “this” lichen here. How about replacing “this” with “L. brialmontii” here?

We rejected ´this´ and replaced by L. brialmontii´.           Ref: line 389-390

L 476-7, Since the ROS production in the lichen thallus is speculation whereas the effect of ROS on PSII is generally observed phenomena, I propose to change these sentences as follows:

During..... thallus, excess ROS formation may occur in PSII, ..... High levels of ROS is harmful not only to PSII....

Former statement was exlusively related to ROS formed in PSII since the next sentence discussed only the PSII limitations caused by ROS in PSII. However, you are right, ROS might be formed in other parts of primary photosyntehtic machinery as well as in other organels during desication.

In R3 version of the MS, we rephrased the statement and mentioned other cell compartments in which ROS are formed during drought stress in lichens, their photobionts respectively. To our best knowledge, the cellular compartments and ROS formation has not yet been studied in desiccating lichenized algae but several review papers report e.g. peroxisomes and the mitochondria in plants.                                 Ref: lines 406-409

Ll 612-3, What I suggested was to change this sentence to:

“A similar response was found in blackish cyanolichen Leptogium puberulum with similar values (reflectance below 0.1) [33, 71].”

We changed the sentence according to your suggestion.   Ref: lines 482-483
